# Evaluation of the Performance of ACR TI-RADS Also Considering Those Nodules with No Indication of FNAC: A Single-Center Experience

**DOI:** 10.3390/jcm12020398

**Published:** 2023-01-04

**Authors:** Stefano Amendola, Sium Wolde Sellasie, Francesco Pedicini, Massimo Carlini, Giulia Russo, Nicola Ossola, Andrea Leoncini, Flavia Botti, Elena Bonanno, Pierpaolo Trimboli, Luigi Uccioli

**Affiliations:** 1Division of Endocrinology and Diabetes, CTO Andrea Alesini Hospital, Department of Biomedicine and Prevention, University of Rome Tor Vergata, 00133 Rome, Italy; 2Thyroid Endocrine Surgery, Sant’Eugenio Hospital, 00144 Rome, Italy; 3Servizio di Nutrizione Clinica e Dietetica, Ente Ospedaliero Cantonale (EOC), 6500 Bellinzona, Switzerland; 4Servizio di Radiologia e Radiologia Interventistica, Istituto di Imaging Della Svizzera Italiana (IIMSI), Ente Ospedaliero Cantonale (EOC), 6900 Lugano, Switzerland; 5Department of Clinical Sciences and Translational Medicine, University of Rome Tor Vergata, 00133 Rome, Italy; 6UOSD Anatomia Patologica Sant’Eugenio Hospital, 00144 Rome, Italy; 7Department of Experimental Medicine, University of Rome “Tor Vergata”, Via Montpellier, 1, 00133 Rome, Italy; 8Servizio di Endocrinologia e Diabetologia, Ospedale Regionale di Lugano, Ente Ospedaliero Cantonale (EOC), 6900 Lugano, Switzerland; 9Facoltà di Scienze Biomediche, Università Della Svizzera Italiana (USI), 6900 Lugano, Switzerland

**Keywords:** FNAC, ACR TI-RADS, risk of neoplasm, rate of malignancy, “not indicated FNACs”

## Abstract

Background: Several US risk stratification score systems (RSSs) have been developed to standardize a thyroid nodule risk of malignancy. It is still a matter of debate which RSS is the most reliable. The purpose of this study is to evaluate: (1) the concordance between the American College of Radiology TI-RADS (ACR TI-RADS) and fine needle aspiration cytology (FNAC), (2) the cancer rate in the ACR TI-RADS categories, (3) the characteristics of nodules evaluated by FNAC even if not formally indicated according to ACR TI-RADS (‘not indicated FNACs”). Methods: From January 2021 to September 2022, patients attending the Endocrinology Unit of the CTO Hospital of Rome for evaluation of thyroid nodules were included. Results: 830 nodules had negative cytology, belonging to TIR2 and TIR1C. One hundred and thirteen nodules were determined to be suspicious for or consistent with malignancy belonging to TIR3B/TIR4/TIR5. Of this last group, 94% were classified as TR4/TR5 nodules. In total, 87/113 underwent surgery. Among these, 73 had histologically proven cancer, 14 turned out to be benign. “Not indicated FNACs” was 623. Among these, 42 cancers were present. Conclusions: This study confirmed the diagnostic power of ACR TI-RADS. In addition, these data suggest revising the ACR TI-RADS indication to FNAC, especially for TR4.

## 1. Introduction

The thyroid gland may contain one or more lesions, structurally distinct from the surrounding parenchyma, called “nodules”. Thyroid nodules are commonly present in the general population of the world, with prevalence ranging from 16% to 68%, according to different studies and the observed population [1]. In the last few years, the widespread use of ultrasound (US) and other radiological techniques [2] has allowed one to also detect non-palpable thyroid nodules, which, in spite of being asymptomatic, often require further diagnostic process, with thyroid scintigraphy and/or fine needle aspiration cytology (FNAC). The former is a morphological and functional investigation that may identify “hot “ (10%), “warm “ (10%), and “cold” (80%) nodules, according, respectively, to increased, equivalent, or decreased radiotracer uptake in comparison to the remaining thyroid parenchyma. FNAC is important to estimate the risk of neoplasm (RON) and to indicate surgery. It is necessary to identify clinically relevant nodules, which can hide cancer (ranging widely from 1% to 10%, according to the different observed populations) [3,4], or cause compressive symptoms (approximately 5%) or thyroid dysfunction (5%). Fortunately, approximately 90% of total thyroid nodules are benign and 95% are asymptomatic, and remain this way during the follow up [5].

Today, several US features may suggest suspicion of malignancy: nodule composition, echogenicity, shape, margins, and presence of echogenic foci [6]. US examination should also be completed by cervical lymph node scan, (I–VI level) to find suspicious features, such as round shape, increased short axis, absence of the hilum, microcalcifications, cystic components, irregular borders, and increased vascularization [7]. In addition to these, other risk factors for malignancy may be also present in the patient’s medical history: childhood radiation (mainly head and neck and whole-body radiation [8]), exposure to ionizing radiation from fall-out in childhood or adolescence [9], and family history of thyroid cancer or hereditary syndromes that include thyroid cancer (e.g., multiple endocrine neoplasia syndrome type 2, familial adenomatous polyposis, and Cowden’s disease). Moreover, it should be investigated if the nodule(s) is growing rapidly or if it is causing hoarseness or other disturbances.

To standardize the malignancy risk estimation nodules in US, in the last few years, five major different risk stratification score systems (RSSs) have been developed: American College of Radiology Thyroid Imaging Reporting and Data System (ACR TI-RADS) [10], Korean Thyroid Imaging Reporting and Data System (K TI-RADS) [11], European Thyroid Imaging Reporting and Data System (EU TI-RADS) [12], AACE/ACE/AME [13], and ATA [14]; these last two are the only two included in clinical guidelines, while the others remain as radiological recommendations.

Several biases can affect the results of this specific literature. First, most studies used only cytology as a reference standard, often excluding less frequent cancers (i.e., follicular and medullary) [15,16]. Second, the largest part of these studies included only series of nodules that underwent FNAC; poor or no information was described for those lesions without indications to FNAC according to RSS. In addition, it is still a matter of debate whether RSSs are fully reliable, and what the most reliable RSS is, if any [17,18]. Even if ACR TI-RADS is the most popular RSS [19], its extent is unknown in the clinical practice. Finally, the potential burden of nodules in which FNAC is not indicated according to RSS remains unexplored. 

Based on the above significant limitations of the available published data, this study was conceived to evaluate (1) the concordance between ACR TI-RADS, as the most frequently investigated RSS, and the FNAC report; (2) the cancer rate in the categories of ACR TI-RADS; (3) the characteristics of nodules/patients who underwent FNAC during clinical practice even if it was not formally indicated according to ACR TI-RADS; indicating that they did not respect the dimensional criteria to perform FNAC. ACR TI-RADS does not recommend FNAC in nodules that are “benign” (TR1) and “not suspicious” (TR2); FNAC is recommended if the maximum nodule’s diameter is >25 mm for “mildly suspicious” (TR3), >15 mm for “moderately suspicious” (TR4), and >10 mm for “highly suspicious” (TR5) [10].

## 2. Materials and Methods

### 2.1. Institutional Guidelines for the Management of Thyroid Nodules

In the Endocrinology Unit of the CTO Hospital (Rome, Italy), all patients with thyroid nodule(s) were evaluated by US during their clinical examination. After their initial assessment, the patients underwent laboratory tests (mainly TSH-reflex and calcitonin). The definition and FNAC indications followed the ACR TI-RADS classification by default. Additional FNACs have been performed according to other factors, such as operator-based indication or patient anxiety, as expected by the ACR TI-RADS white paper [10], or following indication presented by other health care providers (i.e., endocrinologists, surgeons, otorhinolaryngologists, nuclear medicine specialists, general practitioners), or in the absence of a specific indication from RSSs. Once FNAC was performed, the cytological smears were read and classified according to the Italian consensus [20]. The latter includes seven categories: inadequate (TIR1), cystic (TIR1C), not neoplastic (TIR2), low-risk (TIR3A), and high-risk (TIR3B) indeterminate lesion, suspicious of (TIR4), and consistent with (TIR5) malignancy. The patients were managed according to clinical, US, and FNAC data. All patients included signed the informed consent and privacy forms.

### 2.2. Case Selection

The study period was January 2021 to September 2022. From our outpatient database, patients who underwent both diagnostic workup for thyroid nodules and total or partial thyroidectomy were selected. 

Inclusion criteria were: (I) Age >18 years; (II) detailed preoperative thyroid US examination performed by a skilled and experienced endocrinologist; (III) availability of data on thyroid autoimmunity; (IV) availability of a US-guided FNAC; (V) total or partial thyroidectomy performed in our Thyroid Surgery Unit; (VI) availability of a histological diagnosis in our pathology database. Exclusion criteria were: (I) having performed any of the aforementioned procedures in another clinical setting; (II) “hot nodules” on thyroid scintigraphy not submitted to FNAC.

### 2.3. Measures and Reference Standard

The measures to analyze the reliability of ACR TI-RADS were the following: (1) rate of malignancy (ROM) using histological examination; (2) risk of neoplasm (RON) using FNAC reports. ROM was calculated according to the histological diagnosis of malignancy. RON was defined as negative (−ve) in the presence of the FNAC report as TIR1C and TIR2, and positive (+ve) in the presence of TIR3B, TIR4, and TIR5. Histological examination was the gold standard. In the absence of histology, cytological diagnosis was used as reference to Italian Consensus [20].

### 2.4. Statistical Analysis

All statistical analyses were performed using SPSS software. A *p*-value of <0.05 was taken to indicate a significant difference. Continuous variables are expressed as mean ± standard deviation or as median and interquartile ranges (IQR), when appropriate. Qualitative data are expressed as frequencies. Frequencies were compared by χ^2^-test.

## 3. Results

The demographic and clinical characteristics of the study series are summarised in Table 1. The median age was 58 years, with 80% of patients being women and 10% showing positive thyroid autoimmunity. The median of the largest diameter of the nodules was 15 mm. The distribution of nodules according to ACR TI-RADS showed that TR3 and TR4 had a higher frequency.

During the study period, 1132 biopsies were performed. For the study purpose, nodules with TIR1 and TIR3A (16.7%), due to inconclusive reports by FNAC, were excluded. In total, 943 nodules were included. The cytological diagnosis of included nodules was: TIR1C n. 46 (4.1%), TIR2 n. 781 (69%) TIR3B n. 43 (3.8%), TIR4 n.30 (2.6%), and TIR5 n. 43 (3.8%). Consequently, as defined in the Methods section, there were −ve (TIR1C and TIR2) and +ve (TIR3B, TIR4, TIR5) groups. As presented in Figure 1, when we analyzed the association between ACR TI-RADS and RON, we found 94% of TIR3B/TIR4/TIR5 nodules were ultrasonographically classified as TR4/TR5. The χ^2^-test of the ACR TI-RADS categories and the cytology groups (−ve and +ve) showed a statistically significant difference (*p* < 0.01). The RON of each ACR TI-RADS is detailed in Table 2.

All 113 patients with FNAC suspicious of or consistent with malignancy were referred for surgery; 87/113 underwent lobectomy or total thyroidectomy. Among these, 73 had histologically proven cancer (70 papillary carcinoma, 2 noninvasive follicular thyroid neoplasm with papillary-like nuclear features—NIFTP, 1 thyroid tumor of uncertain malignant potential–UMP). The ACR TI-RADS classification in this group is composed of the following: TR3 n.1 (1.4%), TR4 n.39 (53.4%), TR5 n.33 (45.2%). In the 14 cases of the benign group, we found: TR3 n.1 (7.1%), TR4 n.10 (71.4%), and TR5 n.3 (21.5%). The ROM recorded in ACR TI-RADS is detailed in Table 2.

Finally, according to the third objective of the characteristics of the study, we analyzed the ACR TI-RADS of those nodules which underwent FNAC even if it was not formally indicated by this RSS (that is, the setting generally called “not indicated FNACs”). The total number of “not indicated FNACs” was 623, and among these, 524 were ultrasonographically evaluated as TR3-5. On histological examination, 42 cancers were finally found: 40 papillary carcinoma, 1 tumor of uncertain malignant potential (UMP), and 1 noninvasive follicular thyroid neoplasm with papillary-like nuclear features (NIFTP). TNM [21] is detailed in Table 3. The distribution of ACR TI-RADS categories in this group was: n.1 TR3, n.28 TR4, and n.13 TR5. The median size of this group was 9 mm with an IQR of 8–11 mm.

Lastly, RON and ROM were separate in “indicated FNACs” and “not indicated FNACs” subgroups, according to ACR TI-RADS categories (Table 4). Unfortunately, the two nodules (8%) in the “indicated FNACs” group classified as TR3, with positive cytology (+ve), did not undergo surgery, thus we cannot calculate the ROM in this subgroup. The χ^2^-test did not show a statistically significant difference in RON between the FNACs groups, “not indicated” and “indicated”, according to different ACR TI-RADS (TR3/TR4/TR5) (*p* = 0.09). The χ^2^-test did not show a statistically significant difference in ROM between the FNACs groups of “not indicated” and “indicated” according to different ACR TI-RADS (TR4/TR5) (*p* = 0.25).

## 4. Discussion

The aim of this study has been to evaluate the diagnostic performance of ACR TI-RADS scoring systems that have cytology and histology as reference. Taking into account the former, ACR TI-RADS showed a comparable performance in terms of RON as previously reported [22]. In fact, a significant trend towards an increase of RON was observed in ACR TI-RADS when going from the lower- to the higher-risk categories, in line with what was previously described in the current literature, and, in particular, about the initial ACR TI-RADS white paper [23]. According to our data, ROM using histology as gold standard increased from 50% in TR3 to 92% in TR5. The RON and ROM values in this study are higher than those presented in the literature [24]. However, the peculiarity of our population should be taken into account. First, patients underwent biopsies according to indications given by other clinicians. Second, surgery was indicated only in cases of suspicion of malignancy or malignant cytology. These two aspects can partially explain the higher cancer rate in our series. The ROM value for each ACR TI-RADS category is high and inconsistent with the respective RON. However, this can be justified because only the most suspicious nodules underwent surgery. This might represent a selection bias. The third objective of this study was to quantify the executed ‘not indicated FNACs’, according to the ACR TI-RADS white paper. Analysis of the FNACs subgroups of ‘not indicated’ versus ‘indicated’, according to ACR TI-RADS, did not show a statistically significant difference in terms of RON and ROM. This result confirmed the good performance of ACR TI-RADS regardless of nodule’s size above or below the ACR TI-RADS dimensional threshold. It should be noted that ACR TI-RADS was designed to spare biopsies and reduce overdiagnosis. According to these radiological recommendations, in our series, a very high number of ‘not indicated FNACs’ were found (Table 3), as already reported in another study [25]. However, a rigorous application of ACR TI-RADS recommendations to our series should have resulted in the loss of 42 confirmed cancer diagnoses. In this group, the presence of nine cases of multifocal papillary thyroid cancer and, moreover, eight cases of regional lymph node metastasis are in evidence (N1a/N1b). This result suggests some concerns about the ACR TI-RADS dimensional target criteria for biopsy. In our experience, nodules with suspicious US features for papillary thyroid carcinoma are measured in around 50% of cases less than 1 centimeter. It should be noted that the largest part of cancer among the ‘not indicated FNACs’ group was found in the TR4 category. In this category, FNAC is indicated only when the diameter is above 15 mm.

Even if the ATA Guidelines suggest active surveillance and conservative therapy for size < 1 cm papillary microcarcinoma, we believe that knowing whether a nodule is neoplastic or not could help in organizing the right ‘follow-up’. Therefore, according to our, and other, experiences [26], it could be clinically useful to perform biopsies in US with highly suspicious nodules even if <1 cm.

The strength of our study is the ACR TI-RADS stratification, blinded by the cytological report. The ACR TI-RADS stratification was always ‘pre-biopsy’ defined and performed by the same operator. The main limit of the study is the selection bias, particularly in surgery data, because only nodules suspicious of malignancy or malignant nodules underwent surgery. No patients with benign nodules underwent surgery.

## 5. Conclusions

In conclusion, this study confirmed the diagnostic power of the ACR TI-RADS scoring system to identify the right FNACs in terms of both RON, considering cytology, and ROM, considering histology. However, these data suggest revising the dimensional cut-off point to indicate biopsy, especially for TR4, to avoid missing cancer diagnosis. These data should be useful to improve the decision-making process [27] and implement active surveillance, when appropriate.

## Figures and Tables

**Figure 1 jcm-12-00398-f001:**
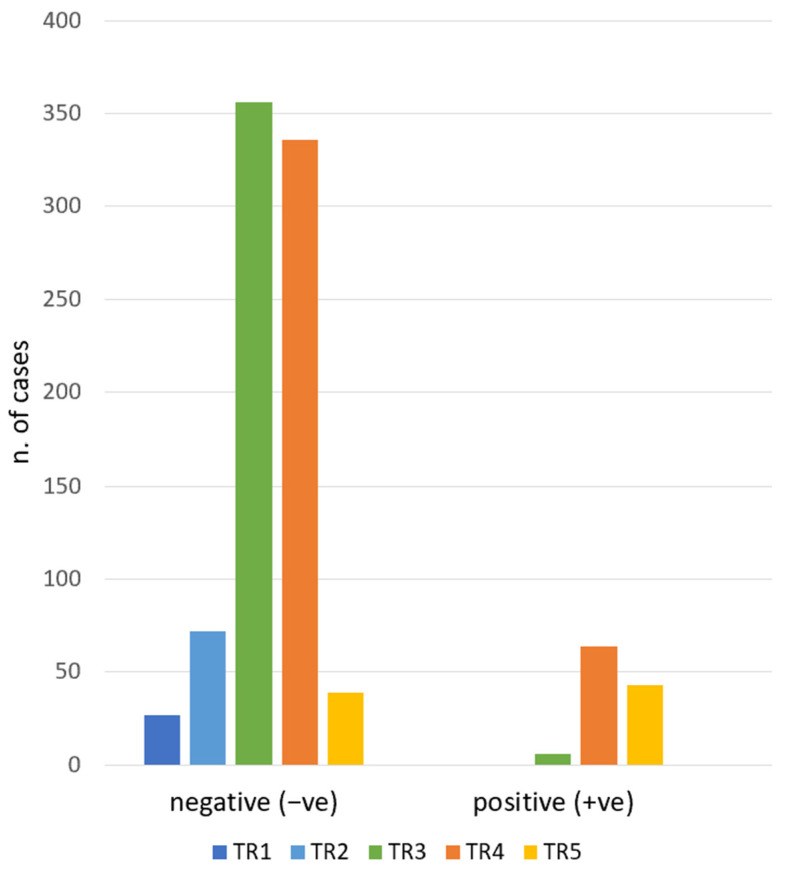
ACR TI-RADS category frequencies according to the cytologic report. TR1: benign. TR2: not suspicious. TR3: mildly suspicious. TR4: moderately suspicious. TR5: highly suspicious. Negative: TIR1C/TIR2. Positive: TIR3B/TIR4/TIR5.

**Table 1 jcm-12-00398-t001:** Demographic and clinical characteristics of the study.

Age	58 (48–66)
Sex	Female 80.2%; Male 19.8%
Positive Thyroid Autoimmunity	10.5%
Maximum diameter (mm)	15 (10–21)
ACR TI-RADS	TR1 n.33 (2.9%)
	TR2 n.232 (20.5%)
	TR3 n.403 (35.6%)
	TR4 n.409 (36.1%)
	TR5 n.55 (4.9%)
Cytology	TIR1 n.147 (13%)
	TIR1C n.46 (4.1%)
	TIR2 n.781 (69%)
	TIR3A n.42 (3.7%)
	TIR3B n.43 (3.8%)
	TIR4 n.30 (2.6%)
	TIR5 n.43 (3.8%)
Histology	Benign n.14 (16%)
	Malignant n.73 (84%)

The frequency of ACR TI-RADS categories and cytologic and histologic reports are expressed as absolute numbers (n.) and in percentage of the total (%). TR1: benign. TR2: not suspicious. TR3: mildly suspicious. TR4: moderately suspicious. TR5: highly suspicious. TIR1: inadequate. TIR1C: cystic. TIR2: benign. TIR3A: indeterminate–low risk. TIR3B: indeterminate–high risk. TIR4: suspicious for malignancy. TIR5: consistent for malignancy.

**Table 2 jcm-12-00398-t002:** RON and ROM according to ACR TI-RADS categories.

	RON (According to FNAC)		ROM (According to Histology)	
TR1	0%	Negative n. 27 (100%) Positive n.0 (0%)	\	\
TR2	0%	Negative n.72 (100%) Positive n.0 (0%)	\	\
TR3	1.7%	Negative n.356 (98.3%) Positive n.6 (1.7%)	50%	Benign n.1 (17%) Malignant n.1 (17%) Active surveillance n.2 (33%) Not available n.2 (33%)
TR4	16%	Negative n.336 (84%) Positive n.64 (16%)	80%	Benign n.10 (15.6%) Malignant n.39 (60.9%) Active surveillance n.13 (20.3%) Not available n.2 (3.2%)
TR5	52%	Negative n.39 (48%) Positive n.43 (52%)	92%	Benign n.3 (7%) Malignant n.33 (76.7%) Active surveillance n.6 (14%) Not available n.1 (2.3%)

RON: risk of neoplasm. ROM: rate of malignancy. TR1: benign. TR2: not suspicious TR3: mildly suspicious. TR4: moderately suspicious. TR5: highly suspicious. Negative: TIR1C/TIR2. Positive: TIR3B/TIR4/TIR5.

**Table 3 jcm-12-00398-t003:** ACR TI-RADS classification of “not indicated FNACs”, cancers found, and TNM.

	“Not Indicated FNACs”	Cancers Founded	TNM
TR1 (No FNAC)	n.27/27 (100% of total TR1)	\	\
TR2 (No FNAC)	n.72/72 (100% of total TR2)	\	\
TR3 (FNAC if >25 mm)	n.258/362 (71.3% of total TR3)	1	Papillary carcinoma n.1: T1b N0
TR4 (FNAC if >15 mm)	n.243/400 (60.7% of total TR4)	28	Papillary carcinoma n.27: T1a N0 n.12 T1b N0 n.5 T1a (m) N0 n.4 T1b (m) N0 n.1 T1a N1a n.2 T1b N1a n.1 T1a N1b n.1 T1b N1b n.1 Uncertain malignant potential (UMP) n.1: T1a N0
TR5 (FNAC if > 10 mm)	n.23/82 (28% of total TR5)	13	Papillary carcinoma n.12: T1a N0 n.6 T1b N0 n.1 T1a (m) N0 n.2 T1a N1a n.1 T1a (m) N1b n. 2 Noninvasive follicular thyroid neoplasm with papillary-like nuclear features (NIFTP) n.1: T1a N0

(T): primary tumor. (N): regional lymph nodes. (M): distant metastasis. (m): multifocal. TR1: benign. TR2: not suspicious. TR3: mildly suspicious. TR4: moderately suspicious. TR5: highly suspicious.

**Table 4 jcm-12-00398-t004:** Analysis of RON and ROM for each ACR TI-RADS category in “not indicated” vs “indicated” FNACs groups.

	“Not Indicated FNACs”		“Indicated FNACs”	
	RON	ROM	RON	ROM
TR1	0%	\	0%	\
TR2	0%	\	0%	\
TR3	1.6%	50%	8%	No data available
TR4	16%	90%	15.3%	61%
TR5	60%	100%	47%	86.9%

RON: risk of neoplasm. ROM: rate of malignancy. TR1: benign. TR2: not suspicious. TR3: mildly suspicious. TR4: moderately suspicious. TR5: highly suspicious.

## Data Availability

Not applicable.

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
