# Peer review of "Evaluation of the Performance of ACR TI-RADS Also Considering Those Nodules with No Indication of FNAC: A Single-Center Experience"

_jcm, 2023, doi:10.3390/jcm12020398_

Round 1
Reviewer 1 Report
Abstract
Line 38 The cited study says up to 68 %
Line 24 and Results line 174 respectively: "Unnecessary FNACs" can be misleading, as they are most commonly defined as FNACs performed in histologically benign nodules. See for example your citation 21. or:
S. J. Yoon et al., ‘Similarities and Differences Between Thyroid Imaging Reporting and Data Systems’, AJR Am. J. Roentgenol., vol. 213, no. 2, pp. W76–W84, Aug. 2019, doi: 10.2214/AJR.18.20510.
S. M. Ha et al., ‘Diagnostic Performance of Practice Guidelines for Thyroid Nodules: Thyroid Nodule Size versus Biopsy Rates’, Radiology, vol. 291, no. 1, pp. 92–99, Apr. 2019, doi: 10.1148/radiol.2019181723.
T. Xu et al., ‘Validation and comparison of three newly-released Thyroid Imaging Reporting and Data Systems for cancer risk determination’, Endocrine, vol. 64, no. 2, pp. 299–307, May 2019, doi: 10.1007/s12020-018-1817-8.
Introduction
Line 39/40: This is not true.
Line 43: What about the role of thyroid scintigraphy? (This question also applies for the Material & Methods section)
Line: 46: 10 % seems overestimated, see for example:
Malignandy rates in thyroid nodules: a long-term cohort study of 17,592 patients. Eur Thyroid J. 2022 Jun 29;11(4):e220027. doi: 10.1530/ETJ-22-0027. Print 2022 Aug 1.PMID: 35635802
Material and Methods
Lines 89-93 and Discussion line 198/199: These "indications" seem partly more than questionable
Results
The results should be shown for TR 3-5 nodules < 1 cm
Correlations between/Performance of TIR and/vs. TR should be calculated/presented.
Table 2: Nodules under "active surveillance" and with unavailable histology cannot be interpreted and should be excluded from statistic analysis.
Table 3: Information on the cancer types (histologic type, TNM, ...) should be provided.
Performance of ACR-TIRADS in nodules with indicated vs. not indicated FNAC should be analysed seperately.
Discussion
Line 209 More information on these 42 cancers should be given.
Author Response
Dear Reviewer, thank you for the comments and the suggestions. I used red in the manuscript to differentiate the text correction after your review.
Abstract
Line 38 The cited study says up to 68 %
Response: I corrected the citation.
Line 24 and Results line 174 respectively: "Unnecessary FNACs" can be misleading, as they are most commonly defined as FNACs performed in histologically benign nodules. See for example your citation 21. or:
- J. Yoonet al., ‘Similarities and Differences Between Thyroid Imaging Reporting and Data Systems’,AJR Am. J. Roentgenol., vol. 213, no. 2, pp. W76–W84, Aug. 2019, doi: 10.2214/AJR.18.20510.
- M. Haet al., ‘Diagnostic Performance of Practice Guidelines for Thyroid Nodules: Thyroid Nodule Size versus Biopsy Rates’,Radiology, vol. 291, no. 1, pp. 92–99, Apr. 2019, doi: 10.1148/radiol.2019181723.
- Xuet al., ‘Validation and comparison of three newly-released Thyroid Imaging Reporting and Data Systems for cancer risk determination’,Endocrine, vol. 64, no. 2, pp. 299–307, May 2019, doi: 10.1007/s12020-018-1817-8
Response: I removed "unnecessary" , I leaved only "not indicated by ACR TI-RADS" in the abstract and in the text, to avoid misleading message.
Introduction
Line 39/40: This is not true.
Response: I removed it from the text.
Line 43: What about the role of thyroid scintigraphy? (This question also applies for the Material & Methods section)
Response: I added that thyroid scintigraphy is a part of thyroid nodule diagnostic work up. I modified it also in the material and methods section.
Line: 46: 10 % seems overestimated, see for example:
Malignandy rates in thyroid nodules: a long-term cohort study of 17,592 patients. Grussendorf M, Ruschenburg I, Brabant G.Eur Thyroid J. 2022 Jun 29;11(4):e220027. doi: 10.1530/ETJ-22-0027. Print 2022 Aug 1.PMID: 35635802
Response: I added it and I cite it ; it's a wide range and depends on population observed.
Material and Methods
Lines 89-93 and Discussion line 198/199: These "indications" seem partly more than questionable
Response: You're right, but as explained in ACR TI-RADS white paper, it might be possible making biopsy for other clinical indications (… the decision to perform FNA should also account for the referring physician’s preference and the patient’s risk factors for thyroid cancer, anxiety, comorbidities, life expectancy, and other relevant considerations…)
Results
The results should be shown for TR 3-5 nodules < 1 cm
Response: I itemized in the text and in the table 3 ACR TI-RADS categories threshold to make biopsies, in order to better explain the TR 3-5 nodules with “not indicated FNACs” (TR3 < 2.5 cm; TR4 < 1.5 cm; TR5 < 1 cm)
Correlations between/Performance of TIR and/vs. TR should be calculated/presented.
Response: I added the chi-square analysis result and the p-value
Table 2: Nodules under "active surveillance" and with unavailable histology cannot be interpreted and should be excluded from statistic analysis.
Response: The Rate of Malignancy is calculated on histological exam (malignant/ not malignant) es. TR3 we found n.1 benign and n.1 malignant, so the rate of malignancy is 50 %. The active surveillance and the other not available histologic report were excluded
Table 3: Information on the cancer types (histologic type, TNM, ...) should be provided.
Response: I added pTNM and histologic type
Performance of ACR-TIRADS in nodules with indicated vs. not indicated FNAC should be analysed seperately.
Response: I created a new explaining table, and I calculated the Risk of Neoplasm and the Rate of Malignancy in the "indicated" and "not indicated" nodules subgroups.
Discussion
Line 209 More information on these 42 cancers should be given.
Response: I detailed these 42 cancers group, as explained before in the results section
Merry Christmas and Happy New Year
Best Regards
Reviewer 2 Report
Wouldn't it be better to write what ACR stands for in abstract section?
Line39: In the XX century ~. What is XX??
In Abstract section, you wrote “Background”, “Methods”, “Conclusions”.
You must write “Results” in this section.
It is better to put a space between the % and the number throughout the manuscript.
Line52-53, 55-59, 104-110
The text is cut and paste, so why not write it in your own words?
Line83, 99, 112, 122
How about removing the spaces, tightening the text, and italicizing?
Line92
Wouldn't it be better to write what ENT stands for?
Line142
Is “Hystology” a mistake of ”Histology” ?
In Table 1, you wrote “Number, % of total” (for example, 2, 9%)? If so, you must write the introduction about what you wrote meant. If you wrote the comma as the decimal point, please rewrite it.
In figure 1, there is an unnatural white space on the right side. Delete it if you don't need it.
Table 2 was difficult to understand. For readers who don't know your abbreviation, the explanation should be neatly written as a footnote.
I understand the importance of research, but it is difficult to understand with the current data and explanations.
Instead of suddenly writing the abbreviated name, please write it with the full name the first time it appears.
It would be better to have a figure that clearly describes your work rather than just giving out data in a table (like charts and illustrations).
Please check the explanation for authors again about how to write references.
Author Response
Dear Reviewer, thank you for the comments and suggestions. I used red to differentiate the text changes.
Wouldn't it be better to write what ACR stands for in abstract section?
Line39: In the XX century ~. What is XX??
Response: I wrote what ACR TI-RADS stand for in the abstract section. I Removed XX century
In Abstract section, you wrote “Background”, “Methods”, “Conclusions”.
You must write “Results” in this section.
Response: I add “results” in the Abstract.
It is better to put a space between the % and the number throughout the manuscript.
Response: I put the space as adviced.
Line52-53, 55-59, 104-110
The text is cut and paste, so why not write it in your own words?
Response: I rewrote the indicated lines in our words, but about inclusion criteria lines it’s very difficult to change because they’re always the same inclusion criteria.
Line83, 99, 112, 122
How about removing the spaces, tightening the text, and italicizing?
Response: I reintroduced the spaces ad adviced
Line92
Wouldn't it be better to write what ENT stands for?
Response: ENT stands for otorhinolaryngologists (Ear, Nose, Throat physician)
Line142
Is “Hystology” a mistake of ”Histology” ?
Response: I apologize for the mistake.
In Table 1, you wrote “Number, % of total” (for example, 2, 9%)? If so, you must write the introduction about what you wrote meant. If you wrote the comma as the decimal point, please rewrite it.
Response: I wrote it in the table 1 introduction (…the frequencies of ACR TI-RADS categories, cytologic and histologic report are expressed in absolute number(n.) and in percentage of the total…)
In figure 1, there is an unnatural white space on the right side. Delete it if you don't need it.
Response: I removed it
Table 2 was difficult to understand. For readers who don't know your abbreviation, the explanation should be neatly written as a footnote.
Response: I introduced the footnote to better explain figures and tables
Merry Christmas and Happy New Year
Best Regards
Reviewer 3 Report
The manuscript has a clinical value, however, it has insufficient data and has to be re-written.
1. There are 3 issues in the aim, however in the conclusion there are not specific answers to those 3 noted issues.
2. It is not clear why authors used ACR TIRADS instead of EU TIRADS or those included in guidelines, and not used Bethesda classification for FNAC but Italian report system? It would be much easier to convert these FNAC reports into Bethesda classification system since most centers use this system. What about ACUS category? Were these FNAC repeated?
3. What exactly ROM means? Risk of malignancy using histological examination? Histological report after the surgery provides clear diagnosis/malignant or not-there is no RISK of malignancy, but malignancy is confirmed or not! This has no sense….
4. Fig 2 is not necessary since it shows the same as Table 3.
Author Response
Dear Reviewer, thank you for the comments and suggestions. I used red to differentiate the text changes.
There are 3 issues in the aim, however in the conclusion there are not specific answers to those 3 noted issues.
Response: The three issue were based on cytology, histology and “ not indicated” FNAC; in the conclusion we reported ACR TI-RADS performance using RON (citology) and ROM(histology). The "not indicated" analysis highlights ACR TI-RADS limits in cancer diagnosis in nodules smaller than the ACR TI-RADS threshold. I modified the text in order to clearly answer to aim issues.
It is not clear why authors used ACR TIRADS instead of EU TIRADS or those included in guidelines, and not used Bethesda classification for FNAC but Italian report system? It would be much easier to convert these FNAC reports into Bethesda classification system since most centers use this system. What about ACUS category? Were these FNAC repeated?
Response: It’s a retrospective study, we continued to use ACR TI-RADS because we had already previous data on nodules classified with ACR TI-RADS. So we did it in order to increase the number of nodules included in the study. We had also specified in the text that even if some metanalysis consider ACR better than the others Risk Score Systems, there is no recent consensus or guidelines confirming this. An analogue answer could be SIAPEC use instead of Betheseda, because we had already cytologic data with SIAPEC and we choosed to continue in this way. AUS fnacs were repated again, if the second time were confirmed AUS they were excluded.
What exactly ROM means? Risk of malignancy using histological examination? Histological report after the surgery provides clear diagnosis/malignant or not-there is no RISK of malignancy, but malignancy is confirmed or not! This has no sense….
Response: you’re right so to avoid misleading message I used the expression of Rate of malignancy (ROM) that means the ratio between the histology confirmed malignancy in ACR TI-RADS category / total histology of that category
Es. Table 2 TR4 ROM: 39 (histology confirmed malignancy) / 49 (total histology of that category, benign+ malignant) = 80%
Fig 2 is not necessary since it shows the same as Table 3.
Response: I removed it as adviced to avoid repetitive messages
Merry Christmas and Happy New Year
Best Regards
Round 2
Reviewer 1 Report
Thank you so much for the corrections. Please have your English revised by a native speaker. The manuscript could be confusing for non-native speaking Europeans.
Author Response
Dear reviewer, thank you for the suggestions, I provided a new manuscript revision by an English native speaker.
Best Regards and Happy New Year
Reviewer 2 Report
Figures must be improved. There is no description of what the vertical axis in Figure 1 is.
The space between the title and subtitle is wide and looks a bit ugly.
In table 2, dose "negative n.356(98,3%)” mean "RON negative patient number is 356 (98.3% of total)" ?
Since this is a study using patient data, do you have permission to use the data?
Author Response
Dear Reviewer, thank you for the suggestions,
Figures must be improved. There is no description of what the vertical axis in Figure 1 is.
The space between the title and subtitle is wide and looks a bit ugly.
In table 2, dose "negative n.356(98,3%)” mean "RON negative patient number is 356 (98.3% of total)" ?
Since this is a study using patient data, do you have permission to use the data?
Response: I added in the Figure 1 axis description; you’re right, they were ugly , I removed the spaces. In table 2, negative n. 356 means 98.3 % of total , but Risk of Neoplasm (RON) is calculated on positive citology cases. All patients included give us permission, I added it in the manuscript (red).
Best Regards and Happy New Year
Reviewer 3 Report
Authors replied to all of my suggestions. The manuscript is acceptable for publication-in my opinion.
Author Response
Dear Reviewer, thank you for the suggestions
Best regards and Happy New Year